# Study on intestinal microbial communities of three different cattle populations on Qinghai-Tibet Plateau

Quji Suolang[1]*, Zhuzha Basang[1], Wangmu Silang[1], Cangjue Nima[1], Qiwen Yang[2], Wa Da[1]

1 Institute of Animal Husbandry and Veterinary Science, Tibet Academy of Agricultural and Animal Husbandry Sciences, Lhasa, China, 2 Key Laboratory of Animal Genetics, Breeding and Reproduction of Shaanxi Province, College of Animal Science and Technology, Northwest A&F University, Xianyang, Shaanxi, China

* 547692283@qq.com

**Data Availability Statement:** The datasets generated from this study are available from the National Center for Biotechnology Information (NCBI)with accession code: PRJNA1099952.

## Abstract

The Tibetan cattle, indispensable·animals on the Qinghai-Tibet Plateau, have become a focal point for the region's economic development. As such, the hybridization of these cattle has been recognized as a pivotal strategy to enhance the local cattle industry. However, research on the gut microbiota of Tibetan hybrid cattle remains scarce. Based on this, we conducted a comparative analysis of the gut microbiota and its functional implications across three distinct cattle populations: two the hybrid cattle populations (Tibetan local cattle × Holstein cattle, TH and Tibetan local cattle × Jersey cattle, TJ) and one the Tibetan locoal cattle population (BL). Bacteroidetes and Firmicutes dominate the gut microbiota across all populations at the phylum level. In addition, the predominant phyla in BL cattle were found to be Cyanobacteria, Verrucomicrobiota, and Actinobacteria, which may be one of the important reasons for the adaptability of Tibetan local cattle to the high-altitude environment of the Qinghai-Tibet Plateau. Further analysis identified specific biomarkers associated with the immune systems of BL cattle, including Bacteroidales_RF16, *Coriobacterium*, and Muribaculaceae. In contrast, TH cattle are primarily dominated by Oscillospiraceae and Clostridia_UCG_014, and TJ cattle are mainly dominated by Christensenellaceae and Gammaproteobacteria. KEGG enrichment analysis revealed that BL and TH cattle showed significant enrichment in the immune system, energy metabolism, and amino acid metabolism-related pathways compared with TJ cattle. Overall, these results suggest that BL and TH cattle demonstrate enhanced adaptability compared to TJ cattle, and indicate that intestinal microbiota of cattle at different altitudes and breeds have diverse structures and functions. Our study presents a new perspective on the role of the microbiome in the hybridization and enhancement of Tibetan cattle.

**Funding:** This work was supported by the demonstration and promotion of healthy cattle breeding technology in Ganggu Village (XZ202201YD0010C); and the project of genomic hypoxic selection signal and gene regulation network of continuous altitude cattle populations (32260823). The funders had no role in study design, data collection and analysis, decision to publish, or preparation of the manuscript.

**Competing interests:** The authors declare that the research was conducted in the absence of any commercial or financial relationships that could be construed as a potential conflict of interest.

## 1. Introduction

The Qinghai-Tibet Plateau, known as the "Roof of the World" is characterized by a cold climate, low oxygen levels, and strong ultraviolet radiation [1], and these harsh and unique environmental conditions have also resulted in the local species developing a strong adaptability to their environment, making the Tibetan Plateau the world's largest high-altitude ecosystems [2]. This region provides valuable raw materials for scholars to research the origin and evolution of species, as well as the sustainable development of natural resources. During the lengthy process of domestication, domestic animals have become stronger evidence of the Qinghai-Tibet Plateau residents' adaption to the high-altitude environment, such as donkeys [3], cattle [4], pig [5, 6], chicken [7, 8], and sheep [9]. Among them, through the study and integration of traditional zoological archaeology identification methods and ancient DNA sequencing technology, it was found that 2500 years ago, the Tibetan Plateau herdsmen had begun to widely raise Tibetan cattle [10]. Bailang cattle (BL) is a local breed of cattle in Bailang County, Tibet Autonomous Region. Widely distributed in Bailang County. BL cattle is an important local livestock, an important source of protein and labor for local herdsmen, and has strong adaptability to high-altitude environments. TH (Tibetan local cattle × Holstein cattle) and TJ (Tibetan local cattle × Jersey Cattle) cattle are hybrid cattle. TH cattle have been bred locally for more than 40 years, while TJ cattle have been raised for about 20 years, and after decades of breeding, the two hybrid cattle have higher milk yield and strong adaptability to high altitudes. At the same time, hybrid improvement is also an important direction for the development of livestock industries by local herdsmen in Tibet in recent decades, which helps to improve the economy and living standards of local herdsmen.

For an extended period, the microorganisms in the gastrointestinal tract have been acknowledged as pivotal determinants in the digestion process, the absorption of nutrients, and the preservation of optimal health in animals. These microscopic inhabitants play a critical role in the intricate balance that sustains the well-being and vitality of their animal hosts [11, 12]. Host and their microbiota have coevolved a mutually beneficial relationship in which the host provides a hospitable environment for the microorganisms and the microbiota provides many advantages for the host, including nutritional benefits and protection from pathogen infection [13], this harmony has also been the result of long-term evolutionary development. Many studies have demonstrated that gut microbes play a vital role in host nutrient uptake and metabolism [14–17], and a variety of bacteria-derived metabolites can affect host endocrine and healthy immune functions [18, 19]. The composition of the gut microbiome in cattle is influenced by a spectrum of factors, which can be broadly categorized into intrinsic and extrinsic elements. Intrinsic factors encompass genetic lineage, such as breed [20] and the transmission of traits through heredity [21]. On the other hand, extrinsic factors encompass the dietary regimen [22], the presence of diseases [23], and the ambient conditions of the environment [24]. Since Tibetan cattle are bred through a combination of captive and free-range methods, their environment plays a crucial role in determining their dietary intake and overall immune health [25]. In addition, the hybrid cattle not only face significant differences in diet but also adapt to the local extreme environment, such as high altitude, cold, and low oxygen levels, which has a huge impact on the composition of their gut microbiome [26]. Therefore, the purpose of this study was to elucidate the classification and functional characteristics of gut microbes in BL cattle, TH cattle, and TJ cattle. This research seeks to offer a theoretical reference for comprehending the makeup and role of gut microbes in hybrid cattle and indigenous cattle populations on the Qinghai-Tibet Plateau. Furthermore, it provides fresh perspectives for examining the adaptability of hybrid cattle in this region.

## 2. Materials and methods

### 2.1 Collection of experimental animals and samples

About 30 fresh fecal samples (about 10 g each) from healthy dry lot feeding cattle were collected from three counties in Tibet Autonomous Region, China: Bailang County (female, n = 10, average annual temperature 5.9 ˚C, altitude 4000m), Shannan City Naidong District (female, n = 10, average annual temperature 5.6 ˚C, altitude3600m) and Yadong County (female, n = 10, average annual temperature 7.7 ˚C, altitude 2800m). The populations were managed with a combination of free-range and captive feeding methods. The diet consisted of a total mixed ration (TMR, TMR provided by Tibet Jiu-feng Feed Co., LTD.), straw, and free outdoor grazing. Fecal samples were collected from the rectal contents of the cattle (Fig 1). Before sampling, the cattle's anus was disinfected and washed with pure water to reduce the risk of foreign substances. Fecal samples were obtained using sterile disposable polyethylene examination gloves; a new glove was utilized for each sample to prevent cross-contamination. Each sample was placed in a labeled 5 mL sterile tube, with each sample stored in a separate sterile tube. The samples were immediately preserved in liquid nitrogen and transported to the laboratory for DNA extraction without delay. Throughout the fecal sample collection process, all samples were collected, handled, and stored with the utmost aseptic technique to prevent any potential contamination. Furthermore, it is noteworthy that none of the cattle had

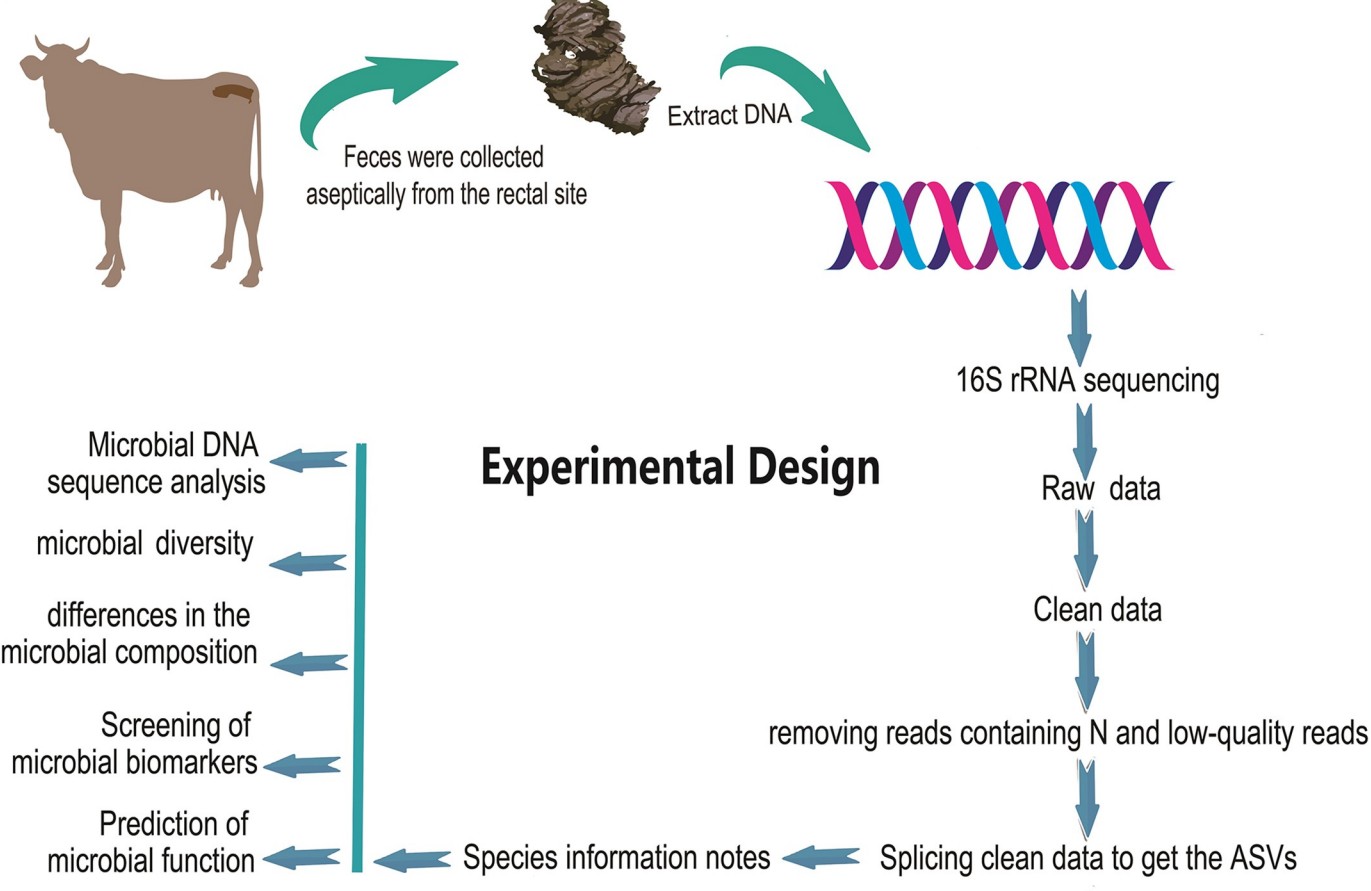

**Fig 1. Experimental design.**

received antibiotic treatment for at least three months prior to sampling, thereby minimizing the influence of such medications on the gut microbiota of the cattle.

During feeding and before the sample, all cattle were regularly examined by a veterinarian to confirm that they were healthy and without any metabolic or gastrointestinal disorder. These animal experiments were approved by the Animal Welfare Committee of Northwest A&F University (FAPWC-NWAFU, protocol number NWAFAC1008), and all procedures were conducted in accordance with the guidelines of the China Animal Protection Association.

## 2.2 Microbial DNA extraction and 16S rRNA gene sequencing

In this study (Fig 1), fecal DNA was extracted using the magnetic bead method following the manufacturer's protocol. Fecal samples, weighing between 0.3–3 g, were ground into a fine powder under liquid nitrogen to maintain a frozen state. The resulting powder was transferred to a 5 mL centrifuge tube and mixed with 3 mL of tissue lysis solution. After incubation at 65 ˚C for 30 minutes in a water bath, the mixture was centrifuged at 4 ˚C for 10 minutes. The supernatant obtained from centrifugation was added to settling buffer 2 and incubated in a water bath at a low temperature for 5 minutes, followed by another round of centrifugation at a low temperature for 10 minutes. Subsequently, 2 mL of supernatant was extracted and combined with 900 μL of lysate, 900 μL of DNA binding buffer, and 50 μL of magnetic bead buffer. The mixture was gently inverted several times, followed by a 2-second centrifugation and incubation at room temperature for 5 minutes. The magnetic bead precipitation was washed twice with 75% ethanol, air-dried, and then preheated to 50 ˚C using an elution buffer by adding 80–300 μl to the mixture. The mixture was left at room temperature for 5 minutes and then transferred into a 1.5 mL centrifuge tube for further analysis. A certain amount of DNA samples was extracted. Utilizing the PacBio sequencing platform, the DNA sample underwent PCR amplification with universal primers 338F (5′-ACTCCTRCGGGAGGCAGCAG-3′) and 806R (5′-GGACTACCVGGGTATCTAAT-3′). The amplified DNA was processed through damage repair, end-repair, and ligation with adapters to form a dumbbell-shaped library. Enzymatic digestion and size selection using BluePippin were performed to purify the library. Finally, the purified library was sequenced using computer-based methods.

## 2.3 Sequencing data processing and bioinformatics analysis

The sequencing reads were first evaluated for quality using iTools Fqtools fqcheck (v0.25) and filtered with readfq (v1.0). Cleaning reads were obtained by removing reads containing N and low-quality reads (defined as those with consecutive ATCG bases) within a 30bp window (Fig 1). The clean reads underwent splicing with FLASH software to isolate Tags in the highly variable region for microbial analysis. Subsequently, the spliced Tags were clustered into Amplicon Sequence Variants (ASVs, 97% similarity) using USEARCH (v7.0.1090) [27], and the resulting sequences were annotated with species information using RDP classifier software with a threshold of 0.6.

Bioinformatics analyses were performed through online web pages on the online BGI cloud platform. We used Mothur (v1.31.2) [28] and R software (v4.3.1) for analyzing Alpha Diversity (**α diversity is mainly Shannon index and Simpson index**). Beta diversity (*β* diversity), which compares microbial community composition in fecal samples from different cattle populations, was assessed by calculating weighted normalized UniFrac distances (using phylogenetic information and abundance) with QIIME2 [29]. and visualized with non-metric multidimensional scaling (NMDS) of gut microbiota. Relative abundance datasets from various samples were used to analyze microbial community composition and differences in abundance

between communities were determined through linear regression analysis. Variations in abundance between different groups were identified using linear discriminant analysis effect size (LEfSe) (LDA > 4). Additionally, PICRUST2 software was used to predict the functional profile of the microbiota and approximate the overall community function (Fig 1).

## 2.4 Statistical analysis

Kruskal–Wallis tests, and Wilcoxtests, were performed to assess differences in the data between the different groups using GraphPad Prism (v8.0) and SPSS (v26.0). Statistical significance was determined using a threshold of $P < 0.05$, indicating a level of significance within the analysis.

# 3 Results

## 3.1 Microbial DNA sequence analysis of three high-altitude cattle populations

In this study, 30 rectal fecal samples from three cattle populations on the Qinghai-Tibet Plateau were analyzed by 16S rRNA high-throughput sequencing technology. After quality control, a total of 1,697,768 high-quality sequences were obtained from 30 samples, with an average of about **56,592** sequences per sample (S1 Table). Meanwhile, the gradually smoothing species accumulation curves certified that the sequencing quality met the expected standards (Fig 2A & 2B) demonstrating the quality of the sequencing results. Further analysis, employing a 97% sequence similarity threshold, led to the identification of **10,334** amplicon sequence variants (ASVs), with an impressive 1,132 being core ASVs, accounting for **10.95%** of the total ASVs detected. BL cattle exhibited the highest number of ASVs (3,608), followed by TH cattle (3,386) and TJ cattle (3,340) (Fig 2C). Variation in ASV quantities suggests differences in gut microbes among the various cattle populations.

## 3.2 Diversity changes of microbial diversity

Microbial diversity plays a crucial role in understanding the richness and composition of gut flora in cattle. Measures of $\alpha$ diversity and $\beta$ diversity are particularly important in this context. $\alpha$ diversity index, such as the Shannon and Simpson index, provide insights into the richness of intestinal flora. In the study, it was observed that the Shannon index exhibited a significant difference between TH and the other two populations ($P<0.05$). The Simpson index showed a significant difference between TJ cattle and the other two cattle populations ($P<0.05$), indicating variations in microbial evenness. However, no significant difference was noted between TH cattle and BL cattle (Fig 3A & 3B). On the other hand, $\beta$ diversity is utilized to portray differences in microbial structure across various samples. In this study, **NMDS** and Principal coordinates analysis (PCoA) were used to analyze the intestinal microbial structure in the three different cattle populations. The PCoA and NMDS results displayed in Fig 3C & 3D demonstrate that the 30 samples can be categorized into two populations: cattle residing above 3500m altitude (TH and BL cattle) and those below 3000m altitude (TJ cattle). The $\alpha$ and $\beta$ diversity results indicate that different cattle populations and feeding environments affect the structure and composition of intestinal microbes in cattle populations.

## 3.3 Analysis of differences in the gut microbial composition

In this study, we analyzed the gut microbiota of three cattle populations at phylum and genus levels. At the phylum level, the relative abundance of Firmicutes and Bacteroidetes was over 92%, with Firmicutes having a relative abundance of over 76% (BL = **76.51%**, TH = **77.74%**,

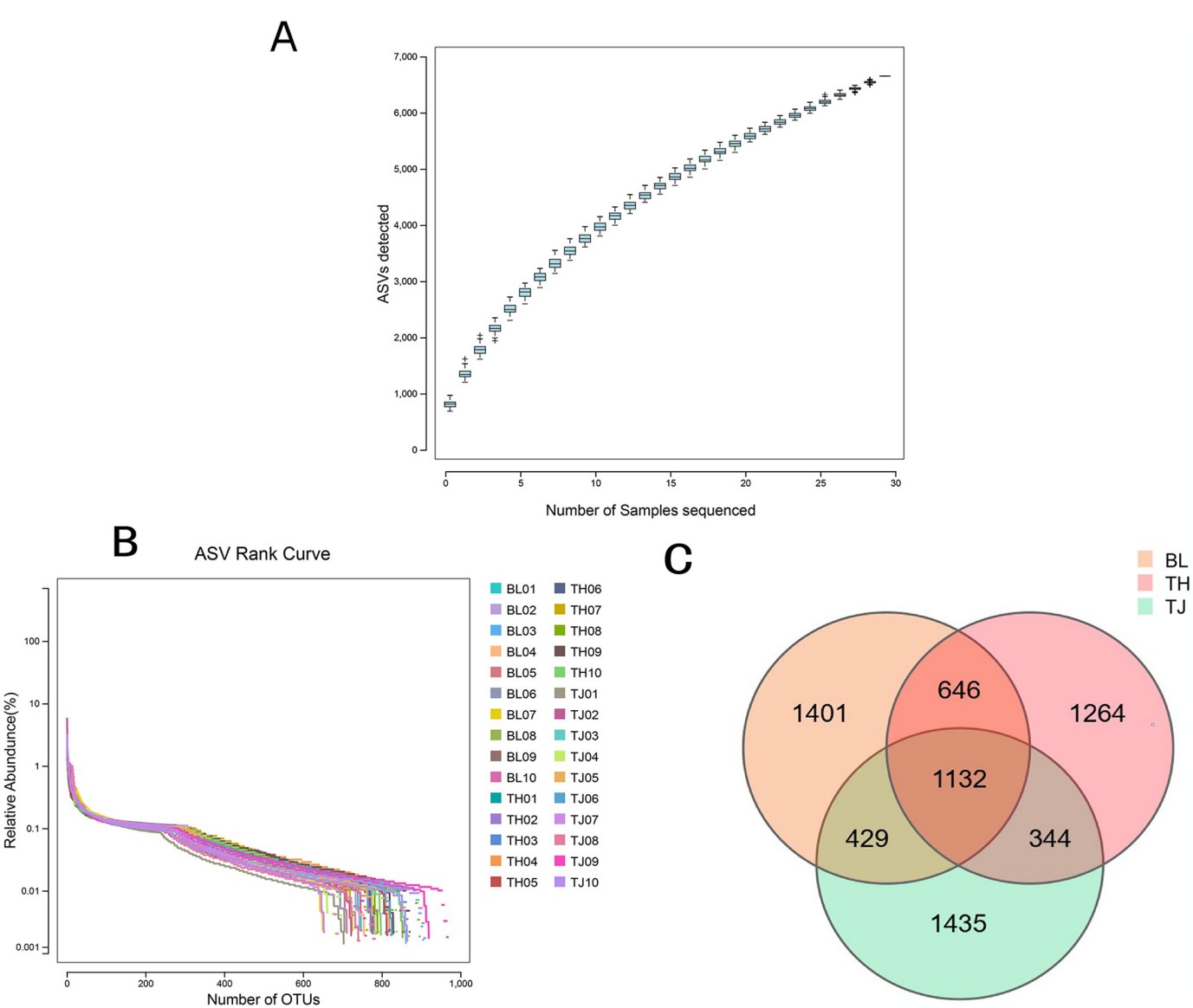

**Fig 2. Statistical analysis of 16S rRNA data.** (A): Intestinal microbial accumulation box type. (B): Rarefaction curves of 30 samples. (C): ASVs distribution of fecal bacterial communities.

TJ = **77.77**%) and Bacteroidetes having a relative abundance of over 16% (BL = **17.13**%, TH = **16.85**%, TJ = **16.89**%) (Fig 4A & 4C and S1 Table). Other phyla, such as Cyanobacteria flora (BL = **2.57**%, TH = **1.40**%, TJ = **0.67**%), Verrucomicrobiota flora (BL = 1.57%, TH = 1.83%, TJ = 1.07%), Proteobacteria flora (BL = **0.43**%, TH = **0.66**%, TJ = **2.01**%) is relatively low in abundance. Among them, Cyanobacteria, Verrucomicrobiota, Proteobacteria, and Actinobacteria have significant differences at the phylum level. At the genus level (Fig 4B & 4D and S1 Table), Christensenellaceae_R-7 (BL = **4.13**%, TH = **5.36**%, TJ = **6.84**%) and *Eubacterium_Coprostanoligenes* (BL = **3.6**4%, TH = **4.39**%, TJ = **4.22**%) was the dominant bacterial genus in the intestinal tracts of the three cattle populations. Christensenellaceae_R-7, *Eubacterium_Coprostanoligenes*, *Phascolarctobacterium*, *Gastranaerophilales*, and *Monoglobus* have significant differences in the genus. These results indicate variations between populations, possibly

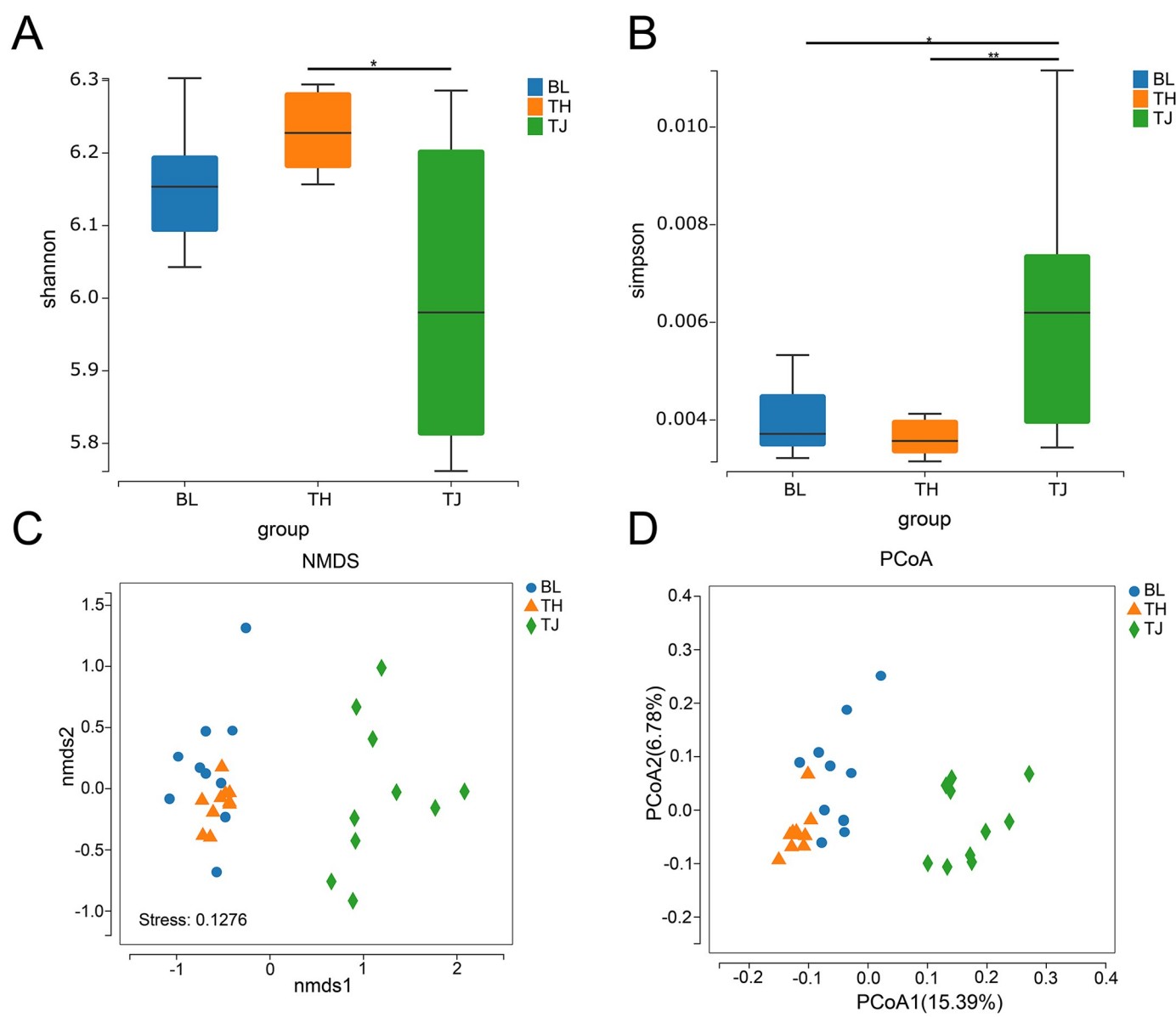

**Fig 3. α and β diversity of gut microbes.** (A): Shannon index. (B): Simpson index. (C)Non-metric multidimensional scale of gut microbiota (NMDS). (D) PCoA distribution of intestinal microbe ASVs.

influenced by the adaptive characteristics of cattle populations and the absorptive capacity of their feed.

## 3.4 Screening of microbial biomarkers from three cattle populations

Linear discriminant analysis (LDA Effect Size, LEfSe, LDA > 2) was employed to identify biomarker species that differentiated intestinal bacterial communities across various cattle populations (Fig 5A & 5B). In BL cattle, the predominant microbial biomarkers included Coriobacteriia, Negativicutes, Vampirivibrionia, Acidaminococcaceae, Selenomonadaceae, Eggerthellaceae, Bacteroidales_RF16, *Coriobacterium*, and Muribaculaceae. The microbial biomarker profile of TH cattle comprised Izemoplasmatales, Clostridia_UCG_014,

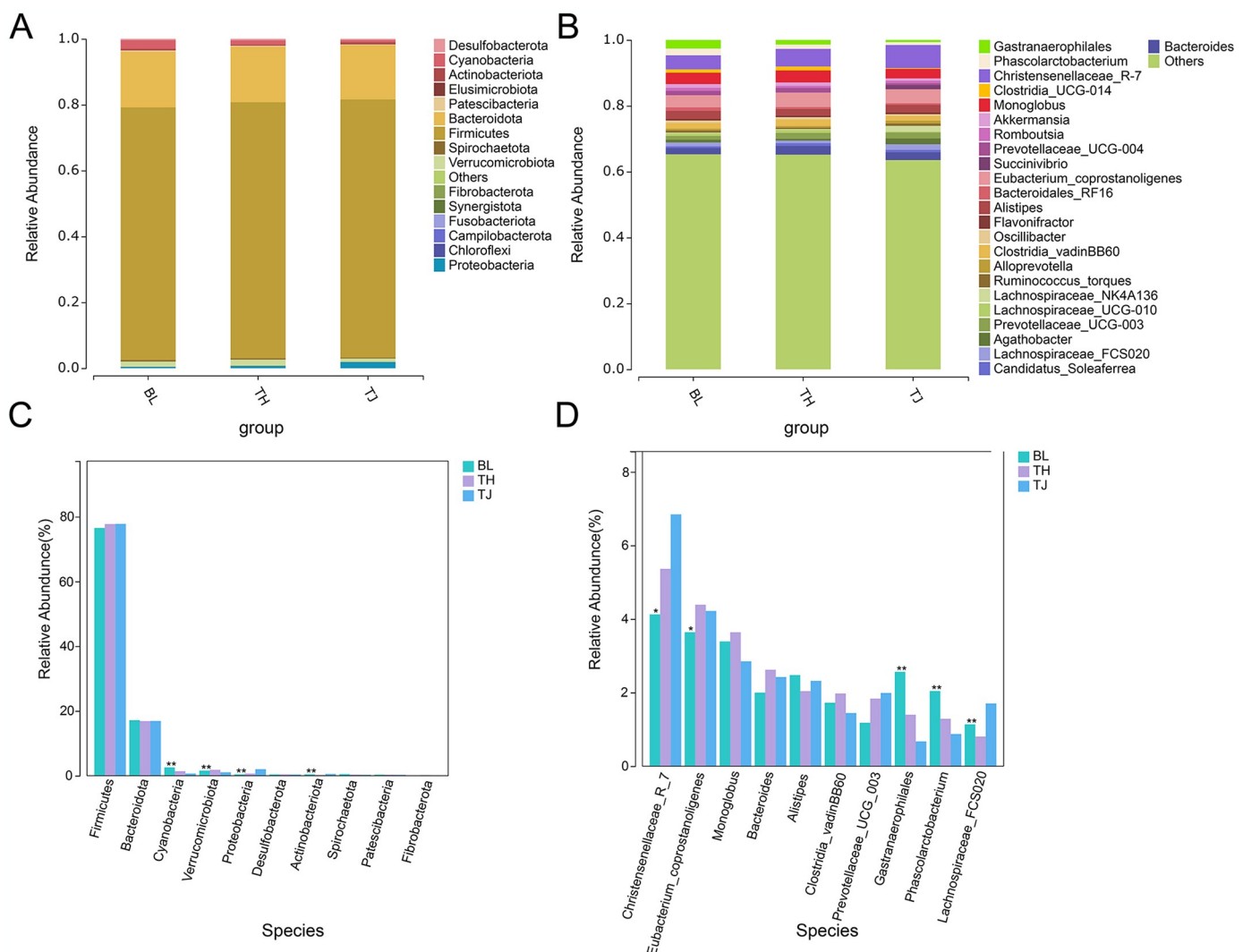

**Fig 4. Composition and differences in the predominant microbiota at the phylum level (A & C) and among genera (B & D) in different populationsare shown.** The abundance of the microbiota is expressed as a percentage. Statistical significance is denoted by *$P < 0.05$ and ***$P < 0.01$.

Oscillospiraceae and Tannerellaceae. Biomarkers specific to TJ cattle were Gammaproteobacteria, Succinivibrionaceae, Christensenellaceae, Streptosporangiaceae, and Pseudonocardiaceae.

### 3.5 Functional predictions of the rectal microbiota in cattle using picrust2

This study aimed to investigate the role of differential intestinal microbiota in cattle populations, utilizing the PICRUSt2 software for prediction and exploration. The analysis conducted at the KEGG secondary level revealed significant functional differences between BL cattle and TH cattle, as illustrated in Fig 6A–6C. The pathways exhibiting significant differences between TH and BL are infectious diseases and bacterial interactions. Conversely, the pathways demonstrating significant variances between BL and TJ cattle encompass the immune system, energy metabolism, translation, nucleotide metabolism, amino acid metabolism, metabolism of terpenoids and polyketides, replication and repair, as well as xenobiotic biodegradation and

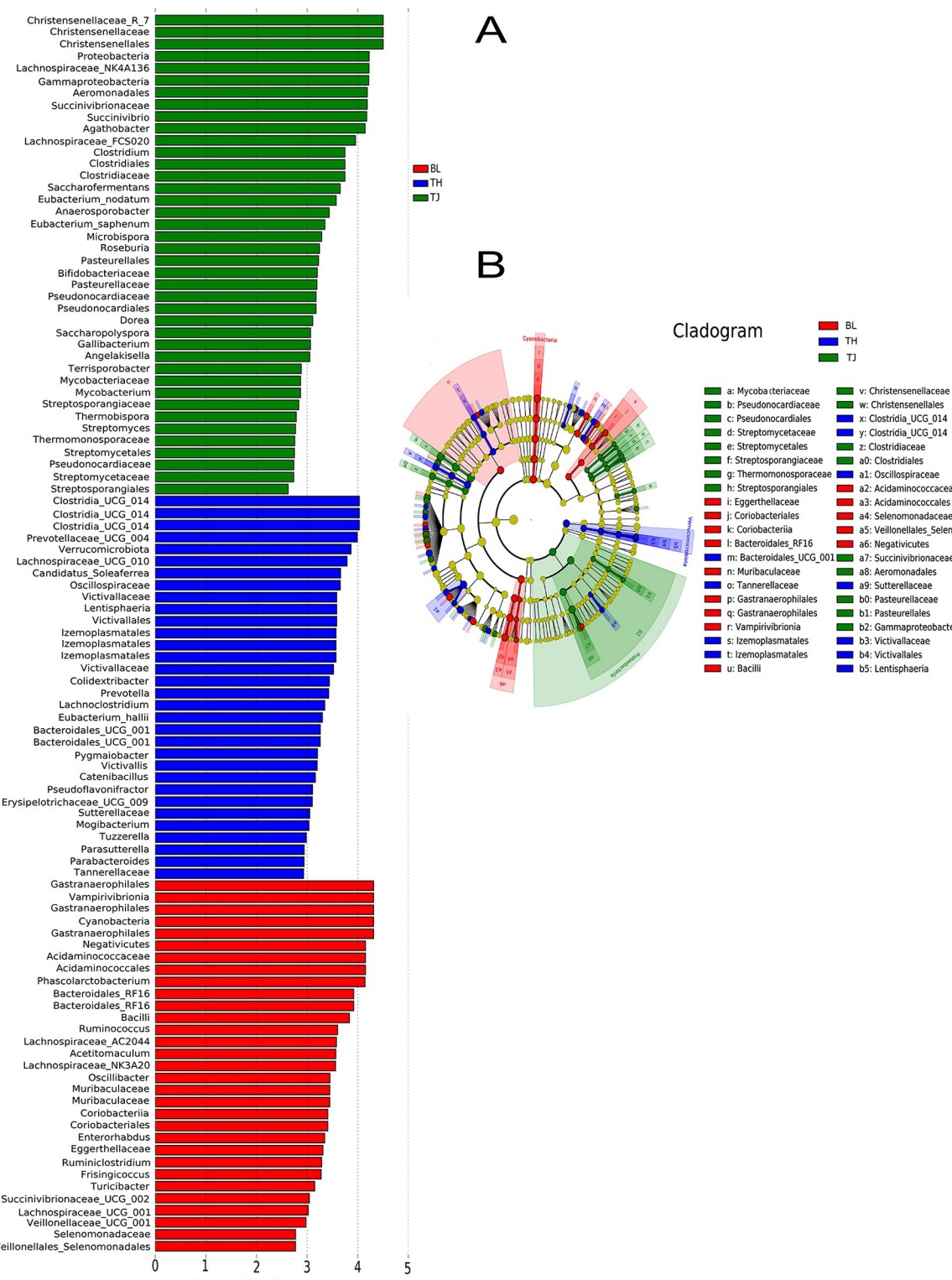

**Fig 5. A. Linear Discriminant Analysis (LDA) and Linear Discriminant Analysis Effect Size (LEfSe) analysis of the intestinal microbial composition in different cattle populations.**

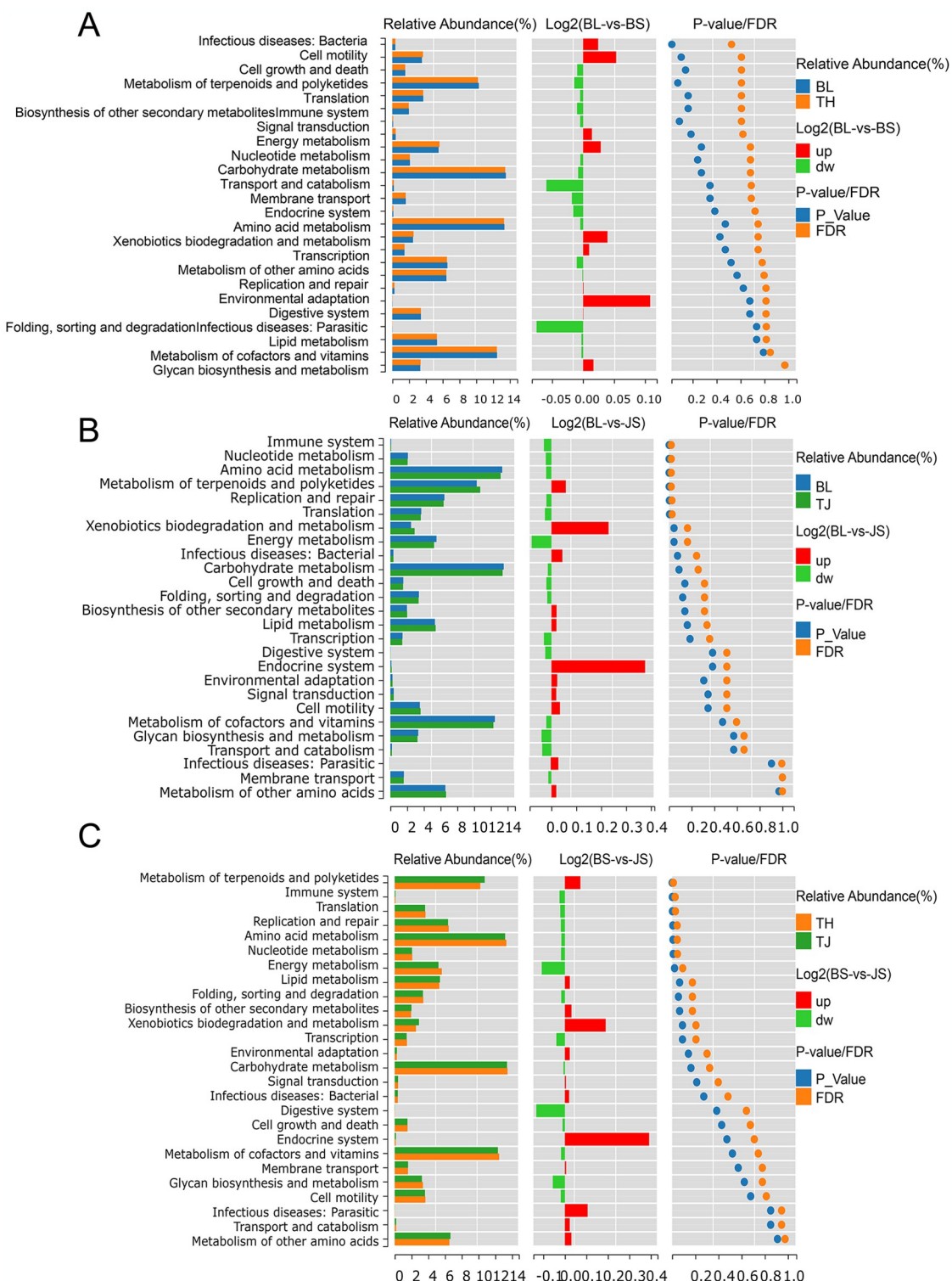

**Fig 6. Based on KEGG intestinal microbial functional enrichment analysis, intestinal microbial enrichment analysis of (A): BL cattle and TH cattle; (B): Enrichment analysis of gut microbes in BL and TJ cattle; (C): Enrichment analysis of gut microbes in TH and TJ cattle.**

metabolism. Furthermore, the significant differences observed between TH and TJ cattle pertained to the immune system, metabolism of terpenoids and polyketides, energy metabolism, translation, **nucleotide and amino acid metabolism**.

## 4. Discussion

Tibetan cattle are an important breed of livestock on the Qinghai-Tibet Plateau, and its high-quality meat, milk, and hides are important resources for local herdsmen [4]. These cattle play a vital role in the livelihood of Tibetan herdsmen, providing a crucial dietary source to help them adapt to the harsh cold climate of the region [30]. Despite their environmental adaptability, local cattle have low production performance, leading to limited economic returns. Therefore, hybridization and the introduction of superior breeds have become pivotal strategies to enhance the economic prospects of local herders. However, paramount consideration is the adaptability of these hybrid cattle and introduced breeds to the local environment, and the transition to a new growth environment can profoundly affect the intestinal microflora of the cattle [31], subsequently influencing their growth, development, and overall economic value. The intestinal microbiota of cattle plays an indispensable role in the digestion of plant fiber, a process that is essential for their nutritional uptake. In addition, each cattle population boasts unique anatomical structures and physiological functions [32], which are finely attuned to their specific environmental conditions and dietary needs.

Bailang County, Yadong County, and Naidong District of Shannan City are all located in the southern part of Tibet, but their altitude has a large difference. The average altitude of Bailang County is 4000 meters, Naidong District of Shannan City is 3600 meters, and Yadong County is 2800 meters. Altitude, environmental humidity, atmospheric pressure, and oxygen content jointly influence the development of intestinal microbiota in animals [33]. Therefore, we analyzed the intestinal flora characteristics of three cattle populations in Tibet. Statistical analysis of ASV data revealed differences in the core flora among the populations, with BL cattle having the highest number of ASVs, while TH and TJ cattle had similar numbers. Microbial alpha diversity analysis results revealed that the Shannon index of BL and TH cattle was significantly higher compared to TJ cattle, indicating greater community diversity. Conversely, BL cattle and TH cattle had significantly lower Simpson index values than TJ cattle, and microbial diversity showed a downward trend with the increase of the Simpson index [34]. The Shannon index and Simpson index demonstrated that the intestinal microbial diversity of BL cattle and TH cattle were significantly higher than that of TJ cattle. The changes can be attributed to the influence of altitude on the environment, which affects plant growth and species diversity, and subsequently impacts the diversity of intestinal microorganisms in herbivores. Research indicates that the species composition of gut microbiota is significantly influenced by altitude. For instance, donkeys [35] at high altitudes exhibit higher gut microbial diversity than those at lower altitudes. Increased intestinal microbial diversity is beneficial for stabilizing the gut ecosystem, enhancing host health [36], mitigating symptoms associated with the anoxic environment of the Qinghai-Tibet Plateau, and improving the adaptability of cattle to extreme cold conditions.

Variability in the microbial composition within cattle populations reflects the diverse microorganism community present. Our study revealed that Bacteroides and Firmicutes are the dominant phyla in the three cattle populations, aligning with the predominant intestinal phyla observed in other ruminants such as sheep [37, 38], elk [39], red deer [40], musk [41] and camel [42]. These phyla play crucial roles in the host's gut, including nutrient digestion and absorption, energy metabolism regulation, and defense against pathogens [16, 43, 44]. The accumulation of Firmicutes and Bacteroides in the intestinal tract aids the host in adapting to the intricate internal environment. At the phylum level, Cyanobacteria, Verrucomicrobiota,

Actinobacteria, and Proteobacteria showed significant variations among the three cattle populations. Notably, the abundance of Cyanobacteria in the gut of BL cattle is significantly greater than in other cattle. Cyanobacteria mainly contribute to photosynthesis-related processes, which has also been confirmed by studies on gut microbes of other high-altitude local animals, such as the Qinghai donkey [35], pika [45], goats [46], and yak [47]. Actinobacteriota is one of the four primary categories of intestinal flora, plays a vital role in maintaining intestinal balance despite its relatively small proportion [48]. The higher levels of Actinobacteriota and Proteobacteria in TJ cattle's intestinal microbiota, compared to BL and TH cattle. It may be related to the abundance of Actinomycetes in the environment in this area, resulting in high actinomycetes content in cattle after grazing, which is consistent with the results of previous studies on yaks [17]. Actinobacteriota is one of the four primary categories of intestinal flora, plays a vital role in maintaining intestinal balance despite its relatively small proportion [18]. The higher levels of Actinobacteriota and Proteobacteria in TJ cattle's intestinal microbiota, compared to BL and TH cattle. It may be related to the abundance of Actinomycetes in the environment in this area, resulting in high actinomycetes content in cattle after grazing, which is consistent with the results of previous studies on yaks [33]. Actinobacteriota contributes to the decomposition of plant-derived Carbohydrates and is involved in the body's immune response, including inflammation and autoimmune reactions [49]. It is also considered as potential probiotics for yaks [33], which can promote the adaptability of TJ cattle to the plateau environment. Interestingly, at the phylum level, there was no significant difference between Bacteroides and firmicutes in the gut microbiota of the three different elevations, possibly due to the feeding of the same TMR (from the same company), which results in minimal differences in the dominant phyla of the microbiotas across populations. At the genus level, *Gastranaerophilales* and *Phascolarctobacterium* are more abundant in BL cattle compared to TH and TJ cattle. *Phascolarctobacterium* produces succinate by degrading crude fibers and provides nutrients to its host as a carbon source for synthesizing short-chain fatty acids (SCFAs). Additionally, *Phascolarctobacterium* has a complete VB12 biosynthesis pathway [50, 51]. Different microorganisms demonstrate varying levels of adaptability across different cattle populations in high-altitude environments.

The LEfSe analysis in this study showed that Bacteroides, Actinomycetes, Bacteroidales_RF16, and Muribaculaceae were microbial markers in the gut of BL cattle. Bacteroides play a crucial role in degrading carbohydrates and providing 10–15% of the host's food energy. Moreover, Bacteroideaceae possesses an efficient degradation system for plant polysaccharides, leading to the production of significant amounts of SCFAs. Studies have shown that its involvement in polysaccharide transport and utilization within the intestinal flora, with approximately 20% of its genome dedicated to sugar decomposition [52]. Additionally, Muribaculaceae has been identified as a key predictor of SCFA concentration in the gut of mice [53]. This microbial group is responsible for the production of succinic acid, acetate, and propionate from specific polysaccharides, including plant glycans, host glycans, and alpha-glucans [54, 55], and this has also been consistently studied in the gastrointestinal tract of yaks [56]. These microbes likely play a role in the long-term adaptation of cattle in the Qinghai-Tibet Plateau. Clostridia_UCG_014 produces SCFAs like propionate and butyrate, which play a role in maintaining the stability of gut microbes [57]. Moreover, previous studies have shown that Oscillospiraceae has a significant positive correlation with the ability to resist parasitic infection, and can improve the host resistance of yaks to parasites at high altitudes and promote intestinal health [58]. Christensenellaceae, known for efficient sugar fermentation, breaks down glucose into acetic acid and butyric acid, as well as plant fibers into volatile fatty acids, crucial for maintaining intestinal balance and healthy gastrointestinal development in ruminants [59, 60]. Christensenellaceae also can secrete $\alpha$-arabinosidase, $\beta$-galactosidase and $\beta$-glucosidase, which are also closely related to feed efficiency [61], and the study found that is vital

for yaks to adapt to the harsh environment of the Tibetan Plateau [56]. Overall, these microbial communities can help alleviate gastrointestinal discomfort resulting from local grass consumption, assisting cattle in adapting to high-altitude environments.

Enrichment analysis using the KEGG database was conducted to investigate the functional roles of intestinal microbiota in distinct cattle populations. BL cattle of the Tibetan Plateau have demonstrated remarkable adaptability to the challenging high-altitude environment characterized by low oxygen levels, and Compared with TJ cattle, BL cattle show a higher enrichment in pathways related to immunity, energy metabolism, DNA replication, repair, and translation. In recent years, studies have found that intestinal tissue is not only the main site for nutrient absorption in humans and animals but also functions as the main immune organ [62, 63], which has also been observed in other high-altitude native species [64]. This enhanced adaptability is likely attributed to their prolonged consumption of local grasses on the Tibetan Plateau, resulting in specific alterations in their intestinal microbiota. These adaptations may be the result of their long-term consumption of local grass on the Tibetan Plateau, leading to specific changes in their intestinal microbiota, which was also confirmed in studies of Tibetan antelopes, Tibetan wild asses, Tibetan sheep, and wild plateau pikas [45, 64]. This further highlights BL cattle's remarkable adaptability to the plateau environment at a microbial level. Both TH and TJ cattle exhibit enrichment in pathways associated with immune system functions, energy metabolism, DNA replication and repair, amino acid metabolism, nucleotide metabolism, and the excretory system, where DNA repair has been demonstrated in adaptations in many other animals at this altitude [65]. In addition, studies have shown that high-altitude extreme environments can improve gut-associated immune systems [66]. In short, the hybrid cattle of different altitudes and different breeds have different adaptability to high altitudes, and the native cattle of Tibet have stronger adaptability to high altitudes than the hybrid cattle.

## 5. Conclusions

In summary, our research utilized high-throughput 16S rRNA microbial sequencing and PICRUSt2 to investigate intestinal microbial functions across various cattle populations. significant variations in microbial abundance and biomarkers among different breeds, underscoring the substantial influence of breed on the intestinal microbiome's composition. BL cattle demonstrated robust adaptability to high-altitude environments, likely due to prolonged evolutionary processes. Interestingly, while both TH and TJ cattle are hybrid breeds, the TH cattle displayed enhanced adaptability to high altitudes in terms of their gut microbial profiles compared to TJ cattle. This difference may be attributed to TH cattle being raised at an altitude of 3,500 meters for over 40 years, undergoing extensive breeding practices, and being selectively bred by multiple breeders, leading to a heightened adaptability to high-altitude environments. Additionally, parental selection may also influence this adaptability, a focal point for our future research endeavors. These findings lay the groundwork for improving Tibetan cattle populations at the microbial level, suggesting that hybrid cattle are a viable option for cattle breeding on the Qinghai-Tibet Plateau. Our study reveals specific changes in intestinal microbes across different cattle herds on the Tibetan Plateau and offers theoretical insights for the breeding of hybrid cattle in this unique environment.

## Supporting information

**S1 Table. Abundance of microbial taxa in the gut.** The list of bacterial communities relative abundance at phylum and genus of every samples and ASV numbers of samples.
(XLSX)

## Acknowledgments

The authors would like to Thanks to the herdsmen of Bailang County, Shannan City Naidong District and Yadong County in the Tibetan Autonomous Region of Tibet for providing cattle. Thanks to Tibet Jiu-feng Feed LTD for their support, as well as to **Northwest Agriculture and Forestry University** (NWAFU) for the resources.

## Author Contributions

**Conceptualization:** Quji Suolang, Zhuzha Basang.

**Data curation:** Qiwen Yang.

**Formal analysis:** Cangjue Nima, Qiwen Yang.

**Funding acquisition:** Quji Suolang.

**Investigation:** Wangmu Silang, Cangjue Nima, Qiwen Yang.

**Methodology:** Quji Suolang, Wa Da.

**Project administration:** Quji Suolang.

**Resources:** Wa Da.

**Software:** Zhuzha Basang, Wangmu Silang, Qiwen Yang.

**Supervision:** Cangjue Nima.

**Validation:** Zhuzha Basang.

**Visualization:** Quji Suolang, Wa Da.

**Writing – original draft:** Quji Suolang.

**Writing – review & editing:** Quji Suolang.

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
