## [Decision Letter · Decision Letter 0]

13 Sep 2024

PONE-D-24-36513Study on intestinal microbial communities of three different cattle populations on Qinghai-Tibet Plateau PLOS ONE

Dear Dr. Suolang,

Thank you for submitting your manuscript to PLOS ONE. After careful consideration, we feel that it has merit but does not fully meet PLOS ONE’s publication criteria as it currently stands. Therefore, we invite you to submit a revised version of the manuscript that addresses the points raised during the review process.

We look forward to receiving your revised manuscript.

Kind regards,

Sayed Haidar Abbas Raza

Academic Editor

PLOS ONE

Journal Requirements:

2. We suggest you thoroughly copyedit your manuscript for language usage, spelling, and grammar. If you do not know anyone who can help you do this, you may wish to consider employing a professional scientific editing service. The American Journal Experts (AJE) (https://www.aje.com/) is one such service that has extensive experience helping authors meet PLOS guidelines and can provide language editing, translation, manuscript formatting, and figure formatting to ensure your manuscript meets our submission guidelines. Please note that having the manuscript copyedited by AJE or any other editing services does not guarantee selection for peer review or acceptance for publication. Upon resubmission, please provide the following: ● The name of the colleague or the details of the professional service that edited your manuscript ● A copy of your manuscript showing your changes by either highlighting them or using track changes (uploaded as a *supporting information* file) ● A clean copy of the edited manuscript (uploaded as the new *manuscript* file)

3. In your Methods section, please provide additional information regarding the permits you obtained for the work. Please ensure you have included the full name of the authority that approved the field site access and, if no permits were required, a brief statement explaining why."""

Jasmine G (Newgen) 3 Sept 2024: ***Straive, at PRTC, please send the following request and do not ping with follow up: "In your Methods section, please provide additional details regarding participant consent from the owners of the animals. In the ethics statement in the Methods and online submission information, please ensure that you have specified (1) whether consent was informed and (2) what type you obtained (for instance, written or verbal). If the need for consent was waived by the ethics committee, please include this information.

4.  Thank you for stating the following financial disclosure: “This work was supported by the demonstration and promotion of healthy cattle breeding technology in Ganggu Village (SD2019XM008); and the project of genomic hypoxic selection signal and gene regulation network of continuous altitude cattle populations (32260823).”

6. PLOS requires an ORCID iD for the corresponding author in Editorial Manager on papers submitted after December 6th, 2016. Please ensure that you have an ORCID iD and that it is validated in Editorial Manager. To do this, go to ‘Update my Information’ (in the upper left-hand corner of the main menu), and click on the Fetch/Validate link next to the ORCID field. This will take you to the ORCID site and allow you to create a new iD or authenticate a pre-existing iD in Editorial Manager.

Reviewers' comments:

Reviewer's Responses to Questions

**Comments to the Author**

1. Is the manuscript technically sound, and do the data support the conclusions?

Reviewer #1: Yes

Reviewer #2: Yes

2. Has the statistical analysis been performed appropriately and rigorously? 

Reviewer #1: N/A

Reviewer #2: Yes

3. Have the authors made all data underlying the findings in their manuscript fully available?

Reviewer #1: Yes

Reviewer #2: Yes

4. Is the manuscript presented in an intelligible fashion and written in standard English?

Reviewer #1: No

Reviewer #2: Yes

5. Review Comments to the Author

Reviewer #1: Title and Abstract:

“In the: Study on intestinal microbial communities of three different cattle populations on Qinghai-Tibet Plateau” following are some recommendations.

Abstract:

The resoult (line 16) should be spell checked. Revise whole manuscript for such type of mistakes.

Introduction:

local species developing a strong adaptability to their environment. Despite the harsh environment, The Tibetan Plateau (The should be the).

The practical applications of study are missing. The data is not fully supported by recent literature.

Materials and Methods:

• Experimental design can be represented in tabulated or pictorial form

• Sample collection and transportation procedures are not justified.

Results:

Scientific nomenclature (e.g. use of italics) should be followed throughout the manuscript. Check the manuscript for this carefully.

Discussion:

The latest Literature should be followed if possible. The discussion doesn’t seem to properly justify the results. There seems to be a lack of in-depth discussion. This portion needs to be revised. ‘some more opposing references can also be added to strengthen this portion

Overall Impression:

Should be revised focusing on grammar, the Quality of the English language, and the above-mentioned points.

Weaknesses:

Need to be much improved. Overall, I consider that this Paper needs moderate Revision and should be improved by an expert in the given field.

Reviewer #2: The manuscript " Study on intestinal microbial communities of three different cattle populations on Qinghai-Tibet Plateau " talks about the study of the intestinal microbial communities of cattle. The authors studied 3 different cattle population including one native and two hybrid populations. The two hybrid populations include Tibetan local cattle × Holstein cattle, TH and Tibetan local cattle × Jersey cattle, TJ and the native population is BL. They found that at the phylum level, the main microbial groups in the gut of these cattle populations were Bacteroidetes and Firmicutes. The authors did find several differences in the biomarkers of different microorganisms between the 3 cattle population. They found that Bacteroidales_RF16, Coriobacterium, and Muribaculaceae are associated with the immune system of BL cattle. The biomarkers in the intestines of TH cattle were mainly Oscillospiraceae and Clostridia UCG_014, while the biomarkers in the intestines of TJ cattle were Christensenellaceae and Gammaproteobacteria. This study is important because cross breeding of cattle can lead to better quality of cattle which are able to survive in native enviroment. This study suggested that BL cattle and TH cattle exhibit superior adaptability compared to TJ cattle, and that intestinal flora of cattle with different altitudes and breeds had different structures and functions. The manuscript is well written but I have a few concerns

Major Concern

1. The manuscript lacks figure legends and without the figure legends it is difficult to interpret the results. Authors need to provide the figure legends and also label the figures completely.

2. The result section needs better headings, the heading are quite similar and at times confusing .

3. The difference in the intestine microbial population at the phylum level is very less for the abundant phyla. The authors should clearly state this as the finding is buried in the text. Firmicutes is over 76% (BL = 76.71, TH= 77.85, TJ = 78.44%) and abundance of Bacteroidetes was over 16% (BL =16.93%, TH = 16.81%, TJ = 16.44%).

Minor Concern

1. Line 253 has an error

2. Figure 2 needs better labelling.

6. PLOS authors have the option to publish the peer review history of their article (what does this mean?). If published, this will include your full peer review and any attached files.

Reviewer #1: No

Reviewer #2: No

---

## [Author Response · Author response to Decision Letter 0]

28 Sep 2024

Response letter

(Manuscript ID: PONE-D-24-36513R1)

Dear PLOS ONE Editor and Reviewer,

Thank for your review comments on our manuscript entitled “Study on intestinal microbial communities of three different cattle populations on Qinghai-Tibet Plateau” (Manuscript ID: PONE-D-24-36513R1). These comments are all valuable and very helpful for improving our manuscript. We have made the corrections. We are hoping that this will meet the approval criteria. We use a change tracking mode to show what needs to be changed the main corrections in the manuscript and the responses to comments are as follows.

Editorial modification

We've checked your submission and before we can proceed, we need you to address the following issues:

Question 1

Thank you for stating the following financial disclosure: “This work was supported by the demonstration and promotion of healthy cattle breeding technology in Ganggu Village (SD2019XM008); and the project of genomic hypoxic selection signal and gene regulation network of continuous altitude cattle populations (32260823).”

Answer 1

Thanks for your valuable comments, we have added a note to the funders in the "Additional Information" document that "funders have no role in the study design, data collection, analysis, publication decisions or manuscript preparation".

Question 2

PLOS requires an ORCID iD for the corresponding author in Editorial Manager on papers submitted after December 6th, 2016. Please ensure that you have an ORCID iD and that it is validated in Editorial Manager. To do this, go to ‘Update my Information’ (in the upper left-hand corner of the main menu), and click on the Fetch/Validate link next to the ORCID field. This will take you to the ORCID site and allow you to create a new ID or authenticate a pre-existing ID in Editorial Manager.

Answer 2

According to your suggestion, we have updated the personal information

Question 3

Please amend the title either on the online submission form or in your so that they are identical.

Answer 3

Thanks to your valuable comments, we have made consistent changes to the topic

Reviewer 1

Comments to the Author:

In the: Study on intestinal microbial communities of three different cattle populations on Qinghai-Tibet Plateau”following are some recommendations.

Question 1

Abstract:The resoult (linee 16) should be spell checked. Revise whole manuscript for such type of mistakes. Introduction:local species developing a strong adaptability to their environment. Despite the harsh environment, The Tibetan Plateau (The should be the). The practical applications of study are missing. The data is not fully supported by recent literature.

Answer 1

Thanks to the reviewer's valuable comments, we have made modifications. we have corrected the spelling throughout the entire article. Furthermore, we have incorporated recent literature and additional descriptions to enhance our discussion on the adaptability of native species from the Tibetan Plateau to high-altitude environments. The following references have been added: [3], [4], [5], [8], [9], [10], [17], [31], [33], [38], [40], [46], [48], [51], [55], [61], [62], and [63]. Specific content related to these references can be found in the manuscript at the following locations: lines 38-41, line 197, line 205, lines 222-223, line 230, line 231, line 248, line 259, line 267, and line.

Question 2

1. Materials and Methods: Experimental design can be represented in tabulated or pictorial form; Sample collection and transportation procedures are not justified.

Answer 2

Thanks for your comments, we have made modifications according to the requirements to further improve the information of the samples, and we have shown the experimental design in the form of pictures, as shown in fig. 1.

Question 3

2. Results: Scientific nomenclature (e.g. use of italics) should be followed throughout the manuscript. Check the manuscript for this carefully.

Answer 3

Thanks to the reviewer's valuable comments, we have made modifications. At the same time, we carefully checked the full text and italicized the names of the level microorganisms in the manuscript, namely Coriobacterium (lines 22 and 174), Eubacterium_coprostanoligenes (lines 165 and 167), Phascolarctobacterium (lines 167, 245-246, and 248), Gastranaerophilales (line167 and 245) and Monoglobus (line 167). 

Question 4

Discussion: The latest Literature should be followed if possible. The discussion doesn’t seem to properly justify the results. There seems to be a lack of in-depth discussion. This portion needs to berevised. some more opposing references can also be added to strengthen this portion.

Answer 4

Thanks to the reviewer's valuable comments, we have made modifications. we have corrected the spelling throughout the entire article. we have incorporated recent literature and additional descriptions to enhance our discussion on the adaptability of native species from the Tibetan Plateau to high-altitude environments. The following references have been added: [31], [33], [38], [40], [46], [48], [51], [55], [61], [62], and [63]. Specific content related to these references can be found in the manuscript at the following locations: lines 38-41, line 197, line 205, lines 222-223, line 230, line 231, line 248, line 259, line 267.

Reviewer 2

Comments to the Author:

The manuscript " Study on intestinal microbial communities of three different cattle populations on Qinghai-Tibet Plateau " talks about the study of the intestinal microbial communities of cattle. The authors studied 3 different cattle population including one native and two hybrid populations. The two hybrid populations include Tibetan local cattle × Holstein cattle, TH and Tibetan local cattle × Jersey cattle, TJ and the native population is BL. They found that at the phylum level, the main microbial groups in the gut of these cattlepopulations were Bacteroidetes and Firmicutes. The authors did find several differences in the biomarkers of different microorganisms between the 3 cattle population. They found that Bacteroidales_RF16, Coriobacterium, and Muribaculaceae are associated with the immune system of BL cattle. The biomarkers in the intestines of TH cattle were mainly Oscillospiraceae and Clostridia UCG_014, while the biomarkers in the intestines of TJ cattle were Christensenellaceae and Gammaproteobacteria. This study is important because cross breeding of cattle can lead to better quality of cattle which are able to survive in native enviroment. This study suggested that BL cattle and TH cattle exhibit superior adaptability compared to TJ cattle, and that intestinal flora of cattle with different altitudes and breeds had different structures and functions. The manuscript is well written but I have a few concerns.

Question 1

The manuscript lacks figure legends and without thefigure legends it is difficult to interpret the results. Authors need to provide the figure legends and label the figures completely.

Answer 1

Thank the reviewers for their valuable comments, and we are very sorry for the mistakes caused by our carelessness. We have modified the drawing notes and legend of the manuscript, as detailed in the manuscript.

Question 2

The result section needs better headings, the heading are quite similar and at times confusing.

Answer 2

We would like to express our gratitude to the reviewers for their valuable comments. In response to their valuable feedback, we have implemented the following implemented: The title "3.2 Analysis of microbial diversity of three high-altitude cattle populations" has been revised to "Diversity changes of three high-altitude cattle populations microbial diversity "(line 143), Replace "3.3 Analysis of differences in the microbial composition of three high-altitude cattle populations" with "3.3 Analysis of differences in the gut microbial composition(line 156) ", Replace "3.4 Screening of microbial biomarkers from three high-altitude cattle populations" with "3.4 Screening of microbial. biomarkers from three cattle populations "(line 170). Replace "3.5 Prediction of microbial function in three high-altitude cattle populations" with "3.5 Functional predictions of the rectal microbiota in cattle using PICRUSt2 "(line 173).

Question 3

The difference in the intestine microbial population at the phylum level is very less for the abundant phyla. The authors should clearly state this as the finding is buried in the text. Firmicutes is over 76% (BL = 76.71, TH= 77.85, TJ = 78.44%) and abundance of Bacteroidetes was over 16% (BL =16.93%, TH = 16.81%, TJ = 16.44%).

Answer 3

Thanks for the reviewer's valuable comments, we have supplemented and elaborated on our manuscript in accordance with these insights. The reviewer noted, "Interestingly, at the phylum level, there was no significant difference between Bacteroides and Firmicutes in the gut microbiota across the three different elevations. This may be attributed to the uniform feeding of the same Total Mixed Ration (TMR) sourced from the same company, which likely results in minimal variations in the dominant phyla of the microbiotas among the populations (lines 242-245).

Question 4

Minor Concern: Lin 253 has an error and Figure 2 needs better labellineg .

Answer 4

Thank the reviewers for their valuable comments. According to the reviewer's valuable comments, we have made modifications and added tags to Figure 3 (formerly Figure 2). For further details, please refer to the manuscript.

We hope this revised manuscript has addressed your concerns, and look forward to hearing from you.

Yours sincerely,

Quji Suolang.

Institute of Animal Husbandry and Veterinary Science, Tibet Academy of Agricultural and Animal Husbandry Sciences

E-mail:547692283@qq.com

---

## [Decision Letter · Decision Letter 1]

4 Oct 2024

Study on intestinal microbial communities of three different cattle populations on Qinghai-Tibet Plateau

PONE-D-24-36513R1

Dear Dr. Suolang

We’re pleased to inform you that your manuscript has been judged scientifically suitable for publication and will be formally accepted for publication once it meets all outstanding technical requirements.

Kind regards,

Sayed Haidar Abbas Raza

Academic Editor

PLOS ONE

Additional Editor Comments (optional):

I appreciate your efforts to respond to review comments.

Reviewers' comments:

Reviewer's Responses to Questions

**Comments to the Author**

1. If the authors have adequately addressed your comments raised in a previous round of review and you feel that this manuscript is now acceptable for publication, you may indicate that here to bypass the “Comments to the Author” section, enter your conflict of interest statement in the “Confidential to Editor” section, and submit your "Accept" recommendation.

Reviewer #1: All comments have been addressed

Reviewer #2: All comments have been addressed

2. Is the manuscript technically sound, and do the data support the conclusions?

Reviewer #1: Yes

Reviewer #2: Yes

3. Has the statistical analysis been performed appropriately and rigorously? 

Reviewer #1: Yes

Reviewer #2: Yes

4. Have the authors made all data underlying the findings in their manuscript fully available?

Reviewer #1: Yes

Reviewer #2: Yes

5. Is the manuscript presented in an intelligible fashion and written in standard English?

Reviewer #1: Yes

Reviewer #2: Yes

6. Review Comments to the Author

Reviewer #1: I have read the revised manuscript. Authors have significantly improved the manuscript during the revision. They have addressed all of my concerns. I suggest this manuscript is acceptable for the publication.

Reviewer #2: The authors have responded to all the concerns. I recommend acceptance of the manuscript in its revised form.

7. PLOS authors have the option to publish the peer review history of their article (what does this mean?). If published, this will include your full peer review and any attached files.

Reviewer #1: **Yes: **Kun Li

Reviewer #2: No

---

## [Editor Report · Acceptance letter]

15 Nov 2024

PONE-D-24-36513R1 

PLOS ONE

Dear Dr. Suolang, 

I'm pleased to inform you that your manuscript has been deemed suitable for publication in PLOS ONE. Congratulations! Your manuscript is now being handed over to our production team.

Kind regards, 

on behalf of

Dr. Sayed Haidar Abbas Raza 

Academic Editor

PLOS ONE